# Stochastic Optimization for Performative Prediction

**Celestine Mendler-Dünner**\*, **Juan C. Perdomo**\*, **Tijana Zrnic**\*, **Moritz Hardt**†
University of California, Berkeley
{mendler,jcperdomo,tijana.zrnic,hardt}@berkeley.edu

## Abstract

In performative prediction, the choice of a model influences the distribution of future data, typically through actions taken based on the model's predictions. We initiate the study of stochastic optimization for performative prediction. What sets this setting apart from traditional stochastic optimization is the difference between merely updating model parameters and deploying the new model. The latter triggers a shift in the distribution that affects future data, while the former keeps the distribution as is. Assuming smoothness and strong convexity, we prove rates of convergence for both greedily deploying models after each stochastic update (greedy deploy) as well as for taking several updates before redeploying (lazy deploy). In both cases, our bounds smoothly recover the optimal $O(1/k)$ rate as the strength of performativity decreases. Furthermore, they illustrate how depending on the strength of performative effects, there exists a regime where either approach outperforms the other. We experimentally explore the trade-off on both synthetic data and a strategic classification simulator.

## 1 Introduction

Prediction in the social world is often *performative* in that a prediction triggers actions that influence the outcome. A forecast about the spread of a disease, for example, can lead to drastic public health action aimed at deterring the spread of the disease. In hindsight, the forecast might then appear to have been off, but this may largely be due to the actions taken based on it. Performativity arises naturally in consequential statistical decision-making problems in domains ranging from financial markets to online advertising.

Recent work [17] introduced and formalized *performative prediction*, an extension of the classical supervised learning setup whereby the choice of a model can change the data-generating distribution. This perspective leads to an important notion of stability requiring that a model is optimal on the distribution it entails. Stability prevents a certain cat-and-mouse game in which the learner repeatedly updates a model, because it no longer is accurate on the observed data. Prior work established conditions under which stability can be achieved through repeated risk minimization on the full data-generating distribution.

When samples arrive one-by-one over time, however, the learner faces a new challenge compared with traditional stochastic optimization. With every new sample that arrives, the learner has to decide whether to deploy the model, thereby triggering a drift in distribution, or to continue to collect more samples from the same distribution. Never deploying a model avoids distribution shift, but forgoes the possibility of converging to a stable point. Deploying the model too greedily could lead to overwhelming distribution shift that hampers convergence. In fact, it is not even clear that fast convergence to stability is possible at all in an online stochastic setting.

## 1.1 Our contributions

In this work, we initiate the study of stochastic optimization for performative prediction. Our main results are the first convergence guarantees for the stochastic gradient method in performative settings. Previous finite-sample guarantees had an exponential dependence on the dimension.

We distinguish between two natural variants of the stochastic gradient method. One variant, called *greedy deploy*, updates model parameters and deploys the model at every step, after seeing a single example. The other, called *lazy deploy*, updates model parameters on multiple samples before deploying a model. We show that both methods converge to a stable solution. However, which one is preferable depends both on the cost of model deployment and the strength of performativity.

To state our results more precisely we recall the formal setup of performative prediction. In performative prediction, we assume that after deploying a model parameterized by $\theta$, data are drawn from the distribution $\mathcal{D}(\theta)$. The distribution map $\mathcal{D}(\cdot)$ maps model parameters to data-generating distributions.

Given a loss function $\ell(z;\theta)$, a *performatively stable* model $\theta$ satisfies the fixed-point condition,

$$\theta \in \arg\min_{\theta'} \mathop{\mathbb{E}}_{z \sim \mathcal{D}(\theta)} \ell(z;\theta') \,.$$

Performative stability expresses the desideratum that the model $\theta$ minimizes loss on the distribution $\mathcal{D}(\theta)$ that it entails. Once we found a performatively stable model, we therefore have no reason to deviate from it based on the data that we observe.

The stochastic gradient method in this setting operates in a sequence of rounds. In each round $k$, the algorithm starts from a model $\theta_k$ and can choose to perform $n(k)$ stochastic gradient updates where each data point is drawn i.i.d. from the distribution $\mathcal{D}(\theta_k)$. After $n(k)$ stochastic gradient updates, the algorithm deploys the new model parameters $\theta_{k+1}$. Henceforth, the data-generating distribution is $\mathcal{D}(\theta_{k+1})$ and the algorithm proceeds to the next round. For greedy deploy, $n(k) = 1$ for all $k$, whereas for lazy deploy $n(k)$ is a hyperparameter we can choose freely.

To analyze the stochastic gradient method, we import the same assumptions that were used in prior work on performative prediction. Apart from smoothness and strong convexity of the loss function, the main assumption is that the distribution map is sufficiently Lipschitz. This means that a small change to the model parameters (in Euclidean distance) leads to small change in the data-generating distribution (as measured in the Wasserstein metric).

Our first main result shows that under these assumptions, greedy deploy achieves the same convergence rate as the stochastic gradient method in the absence of performativity.

**Theorem 1.1** (Greedy deploy, informal)**.** *If the loss is smooth and strongly convex and the distribution map is sufficiently Lipschitz, greedy deploy converges to performative stability at rate $O(1/k)$, where $k$ is the number of model deployment steps.*

Generally speaking, the Lipschitz parameter has to be smaller than the inverse condition number of the loss function for our bound to guarantee convergence. The exact rate stated in Theorem 3.2 further improves as the Lipschitz constant tends to $0$.

In many realistic scenarios, data are plentiful, but deploying a model in a large production environment is costly. In such a scenario, it makes sense to aim to minimize the number of model deployment steps by updating the model parameters on multiple data points before initiating another model deployment. This is precisely what lazy deploy accomplishes as our next result shows.

**Theorem 1.2** (Lazy deploy, informal)**.** *Under the same assumptions as above, for any $\alpha > 0$, lazy deploy converges to performative stability at rate $O(1/k^\alpha)$ provided that $O(k^{1.1\alpha})$ samples are collected between deployments $k$ and $k + 1$.*

In particular, this shows that any distance from optimality $\delta > 0$ can be achieved with $(1/\delta)^c$ model deployments for an arbitrarily small $c > 0$ at the cost of collecting polynomial in $1/\delta$ many samples.

Our main theorems provide upper bounds on the convergence rate of each method. As such they can only draw an incomplete picture about the relative performance of these methods. Our empirical investigation therefore aims to shed further light on their relative merits. In particular, our experiments show that greedy deploy generally performs better than lazy deploy when the distribution map is very Lipschitz, i.e., the performative effects are small. Conversely, lazy deploy fares better when the

distribution map is less Lipschitz. These observations are consistent with what our theoretical upper bounds suggest.

## 1.2 Related work

Perdomo et al. [17] introduced the performative prediction framework and analyzed algorithms for finding stable points that operate at the population level. While they also analyze some finite-sample extensions of these procedures, their analysis relies on concentration of the empirical distribution to the true distribution in the Wasserstein metric, and hence requires exponential sample complexity. In contrast, our analysis ensures convergence even if the learner collects a single sample at every step.

There has been a long line of work [3, 4, 5, 6, 12] within the learning theory community studying concept drift and learning from drifting distributions. Our results differ from these previous works since in performative prediction, changes in distribution are not a passive feature of the environment, but rather an active consequence of model deployment. This introduces several new considerations, such as the conceptual idea of performative stability, which is the main focus of our investigation.

Our work draws upon ideas from the stochastic convex optimization literature [7, 16, 18, 20, 21, 22]. Relative to these previous studies, our work analyzes the behavior of the stochastic gradient method in performative settings, where the underlying objective changes as a response to model deployment.

Lastly, we can view instances of performative prediction as special cases of reinforcement learning problems with nice structure, such as a Lipschitz mapping from policy parameters to the induced distribution over trajectories (see [17] for further discussion). The variants of the stochastic gradient method we consider can be viewed as policy gradient-like algorithms [1, 10, 23, 24] for this setting.

## 2 Preliminaries

We start by reviewing the core concepts of the framework of performative prediction. Afterwards, we set the stage for our analysis of stochastic algorithms by first considering gradient descent at the population level. In doing so, we highlight some of the fundamental limitations of gradient descent in performative settings.

### 2.1 The framework of performative prediction

Throughout our presentation, we focus on predictive models $f_\theta$ that are parametrized by a vector $\theta \in \Theta \subseteq \mathbb{R}^d$, where the parameter space $\Theta$ is a closed, convex set. The model or classifier, $f_\theta$, maps instances $z \in \mathbb{R}^m$ to predictions $f_\theta(z)$. Typically, we think of $z$ as being a feature, label pair $(x, y)$. We assess the quality of a classifier $f_\theta$ via a loss function $\ell(z; \theta)$.

The key theme in performative prediction is that the choice of deployed model $f_\theta$ influences the future data distribution and hence the expected loss of the classifier $f_\theta$. This behavior is formalized via the notion of a *distribution map* $\mathcal{D}(\cdot)$, which is the key conceptual device of the framework. For every $\theta \in \Theta$, $\mathcal{D}(\theta)$ denotes the distribution over instances $z$ induced by the deployment of $f_\theta$. In this paper, we consider the setting where at each step, the learner observes a single sample $z \sim \mathcal{D}(\theta)$, where $f_\theta$ is the most recently deployed classifier. After having observed this sample, the learner chooses whether to deploy a new model or to leave the distribution as is before collecting the next sample.

We adopt the following Lipschitzness assumption on the distribution map. It captures the idea that if two classifiers make similar predictions, then they also induce similar distributions.

**Definition 1** ($\epsilon$-sensitivity [17]). A distribution map $\mathcal{D}(\cdot)$ is *$\epsilon$-sensitive* if for all $\theta, \theta' \in \Theta$:

$$W_1\big(\mathcal{D}(\theta), \mathcal{D}(\theta')\big) \leq \epsilon \|\theta - \theta'\|_2,$$

where $W_1$ denotes the Wasserstein-1, or earth mover's distance.

The value of $\epsilon$ indicates the strength of performative effects; small $\epsilon$ means that the distribution induced by the model $f_\theta$ is not overly sensitive to the choice of $\theta$, while large $\epsilon$ indicates high sensitivity. As an extreme case, $\epsilon = 0$ implies $\mathcal{D}(\theta) = \mathcal{D}(\theta')$ for all $\theta, \theta' \in \Theta$ and hence there are no performative effects, as in classical supervised learning.

Given how the choice of classifier induces a change in distribution, a naturally appealing property of a predictive model in performative settings is that it achieves minimal risk on the distribution that it induces. This solution concept is referred to as *performative stability*.

**Definition 2** (Performative stability). A classifier $f_{\theta_{\mathrm{PS}}}$ is *peformatively stable* if

$$\theta_{\mathrm{PS}} \in \arg\min_{\theta} \mathbb{E}_{z \sim \mathcal{D}(\theta_{\mathrm{PS}})} \ell(z; \theta).$$

We refer to $\theta_{\mathrm{PS}}$ as being performatively stable, or simply stable, if $f_{\theta_{\mathrm{PS}}}$ is performatively stable.

Performative stability captures an equilibrium state in which a classifier induces a shift in distribution by the environment, yet remains simultaneously optimal for this new distribution. These solutions are referred to as stable since they eliminate the need for retraining. Besides eliminating the need for retraining, performatively stable classifiers were also shown to have nearly optimal predictive power on the distribution they induce. More specifically, stable points approximately minimize the *performative risk*, $\mathrm{PR}(\theta) = \mathbb{E}_{z \sim \mathcal{D}(\theta)} \ell(z; \theta)$, in the case of a strongly convex loss and a reasonably small sensitivity parameter $\epsilon$ (Theorem 4.3, [17]).

To illustrate these abstract concepts, we instantiate a simple traffic prediction example with performative effects which will serve as a running example throughout the paper.

**Example 1** (ETA estimation). *Suppose that each day we want to estimate the duration of a trip on a fixed route from the current weather conditions. Let $x \in \{0, 1\}$ denote a binary indicator of whether the current day is sunny or rainy, and suppose that $\Pr\{x = 1\} = p \in (0, 1)$. Let $f_\theta$ denote the deployed classifier which predicts trip duration $y$ from $x$. Assume $y$ behaves according to the following model:*

$$y = \mu + w \cdot x - \epsilon \cdot (f_\theta(x) - \mu),$$

*where $\mu > 0$ denotes the usual time needed to complete the route on a sunny day, $w > 0$ denotes additional incurred time due to bad weather, and $-\epsilon \cdot (f_\theta(x) - \mu)$ denotes the performative effects, for some $\epsilon \in (0, 1)$. Namely, if the model predicts a faster than usual time to the destination, more people want to take the route, thus worsening the traffic conditions and making $y$ large. If, on the other hand, the model predicts a longer trip, then few people follow the route and $y$ is smaller. Suppose that the model class is all predictors of the form $f_\theta(x) = x\theta_1 + \theta_2$, where $\theta = (\theta_1, \theta_2)$ and $\theta_1 \in (0, w), \theta_2 \in (0, 2\mu)$. It is not hard to see that the distribution map corresponding to this data-generating process is $\epsilon$-sensitive.*

*Assume that we measure predictive performance according to the squared loss, $\ell((x, y); \theta) = \frac{1}{2}(y - \theta_1 x - \theta_2)^2$. Then, a simple calculation reveals that the unique performatively stable classifier, satisfying Definition 2, corresponds to*

$$\theta_{\mathrm{PS}} = \left( \frac{w}{1 + \epsilon}, \ \mu \right).$$

*In fact, one can show that $\theta_{\mathrm{PS}}$ is simultaneously optimal in the sense that it minimizes the performative risk, $\theta_{\mathrm{PS}} = \arg\min_\theta \mathrm{PR}(\theta) = \arg\min_\theta \mathbb{E}_{(x,y) \sim \mathcal{D}(\theta)} \ell((x, y); \theta)$.*

## 2.2 Population-level results

Before analyzing optimization algorithms in stochastic settings, we first consider their behavior at the population level. Throughout our analysis, we make the following assumptions on the loss $\ell(z; \theta)$, which hold for broad classes of objectives. To ease readability, we let $\mathcal{Z} = \cup_{\theta \in \Theta} \mathrm{supp}(\mathcal{D}(\theta))$.

(A1) (*joint smoothness*) A loss function $\ell(z; \theta)$ is $\beta$-jointly smooth if the gradient[3] $\nabla_\theta \ell(z; \theta)$ is $\beta$-Lipschitz in $\theta$ *and* $z$, that is for all $\theta, \theta' \in \Theta$ and $z, z' \in \mathcal{Z}$ it holds that $\|\nabla\ell(z; \theta) - \nabla\ell(z; \theta')\|_2 \leq \beta \|\theta - \theta'\|_2$ and $\|\nabla\ell(z; \theta) - \nabla\ell(z'; \theta)\|_2 \leq \beta \|z - z'\|_2$.

(A2) (*strong convexity*) A loss function $\ell(z; \theta)$ is $\gamma$-strongly convex if for all $\theta, \theta' \in \Theta$ and $z \in \mathcal{Z}$ it holds that $\ell(z; \theta) \geq \ell(z; \theta') + \nabla\ell(z; \theta')^\top (\theta - \theta') + \frac{\gamma}{2} \|\theta - \theta'\|_2^2$. For $\gamma = 0$, this condition is equivalent to convexity.

We will refer to $\frac{\beta}{\gamma}$, where $\beta$ is as in (A1) and $\gamma$ as in (A2), as the condition number.

In this paper we are interested in the convergence of optimization methods to performatively stable classifiers. However, unlike classical risk minimizers in supervised learning, it is not a priori clear that performatively stable classifiers always exist. We thus recall the following fact.

**Fact 2.1** ( [17]). *Assume that the loss is $\beta$-jointly smooth (A1) and $\gamma$-strongly convex (A2). If $\mathcal{D}(\cdot)$ is $\epsilon$-sensitive with $\epsilon < \frac{\gamma}{\beta}$, then there exists a unique performatively stable point $\theta_{\mathrm{PS}} \in \Theta$.*

We note that it is not possible to reduce sensitivity by merely rescaling the problem, while keeping the ratio $\gamma/\beta$ the same; the critical condition $\epsilon\beta/\gamma < 1$ remains unaltered by scaling.[4]

The upper bound $\epsilon < \gamma/\beta$ on the sensitivity parameter is not only crucial for the existence of unique stable points but also for algorithmic convergence. It defines a regime outside which gradient descent is not guaranteed to converge even at the population level. To be more precise, consider *repeated gradient descent* (RGD), defined recursively as

$$\theta_{k+1} = \theta_k - \eta_k \mathop{\mathbb{E}}_{z \sim \mathcal{D}(\theta_k)} [\nabla \ell(z; \theta_k)], \quad k \geq 1, \quad \text{where } \theta_1 \in \Theta \text{ is initialized arbitrarily.}$$

As shown in the following result, RGD need not converge if $\epsilon \geq \frac{\gamma}{\beta}$. Furthermore, a strongly convex loss is necessary to ensure convergence, even if performative effects are arbitrarily weak.

**Proposition 2.2.** *Suppose that the distribution map $\mathcal{D}(\cdot)$ is $\epsilon$-sensitive. Repeated gradient descent can fail to converge in any of the following cases, for any choice of positive step size sequence $\{\eta_k\}$:*

    *(a) The loss is $\beta$-jointly smooth (A1) and convex, but not strongly convex (A2), for any $\beta, \epsilon > 0$.*
    *(b) The loss is $\beta$-jointly smooth (A1) and $\gamma$-strongly convex (A2), but $\epsilon \geq \frac{\gamma}{\beta}$, for any $\frac{\gamma}{\beta} > 0$.*

On the other hand, if $\epsilon < \gamma/\beta$ we prove that RGD converges to a unique performatively stable point at a linear rate. Proposition 2.3 strengthens the corresponding result of Perdomo et al. [17], who showed linear convergence of RGD for $\epsilon < \gamma/(\gamma + \beta)$. Proofs can be found in Appendix C.

**Proposition 2.3.** *Assume that the loss is $\beta$-jointly smooth (A1) and $\gamma$-strongly convex (A2), and suppose that the distribution map $\mathcal{D}(\cdot)$ is $\epsilon$-sensitive. Let $\epsilon < \gamma/\beta$, and suppose that $\theta_{\mathrm{PS}} \in \mathrm{Int}(\Theta)$. Then, repeated gradient descent (RGD) with a constant step size $\eta_k = \frac{\gamma - \epsilon\beta}{2(1+\epsilon^2)\beta^2}$ converges to the stable point $\theta_{\mathrm{PS}}$ at a linear rate, $\|\theta_{k+1} - \theta_{\mathrm{PS}}\|_2 \leq \delta$ for $k \geq 4(1+\epsilon^2)\frac{\beta^2}{(\gamma-\epsilon\beta)^2}\log(\|\theta_1 - \theta_{\mathrm{PS}}\|_2 \,/\, \delta)$.*

Together, these results show that $\gamma/\beta$ is a sharp threshold for the convergence of gradient descent in performative settings, thereby resolving an open problem due to Perdomo et al. [17]. Having characterized the convergence regime of gradient descent, we now move on to presenting our main technical results, focusing on the case of a smooth, strongly convex loss with $\epsilon < \gamma/\beta$.

## 3  Stochastic optimization results

We introduce two variants of the stochastic gradient method for optimization in performative settings (i.e. stochastic gradient descent, SGD), which we refer to as *greedy deploy* and *lazy deploy*. Each method performs a stochastic gradient update to the model parameters at every iteration, however they choose to deploy these updated models at different time intervals.

To analyze these methods, in addition to (A1) and (A2), we make the following assumption which is customary in the stochastic optimization literature [7, 19].

(A3) (*second moment bound*) There exist constants $\sigma^2$ and $L^2$ such that for all $\theta, \theta' \in \Theta$:

$$\mathop{\mathbb{E}}_{z \sim \mathcal{D}(\theta)} \left[ \|\nabla \ell(z; \theta')\|_2^2 \right] \leq \sigma^2 + L^2\|\theta' - G(\theta)\|_2^2, \quad \text{where } G(\theta) \stackrel{\text{def}}{=} \arg\min_{\theta'} \mathop{\mathbb{E}}_{z \sim \mathcal{D}(\theta)} \ell(z; \theta').$$

Given the operator $G(\cdot)$, performative stability can equivalently be expressed as $\theta_{\mathrm{PS}} \in G(\theta_{\mathrm{PS}})$.

| Greedy Deploy | Lazy Deploy |
|---|---|
| **Input:** step size sequence $\{\eta_k\}_{k=1}^{\infty}$<br>Deploy initial classifier $\theta_1 \in \Theta$<br>**For each** $k = 1, 2, \ldots$<br>    – Observe $z^{(k)} \sim \mathcal{D}(\theta_k)$<br>    – Update model parameters:<br>        $\theta_{k+1} = \theta_k - \eta_k \nabla \ell(z^{(k)}; \theta_k)$<br>    – Deploy $\theta_{k+1}$ | **Input:** step size sequence $\{\eta_{k,j}\}_{k,j=1}^{\infty}$<br>Deploy initial classifier $\theta_1 \in \Theta$<br>**For each** $k = 1, 2, \ldots$<br>    – Set $\varphi_{k,1} = \theta_k$<br>    – **For each** $j = 1, \ldots, n(k)$ :<br>      1. Observe $z_j^{(k)} \sim \mathcal{D}(\theta_k)$<br>      2. Update model parameters:<br>        $\varphi_{k,j+1} = \varphi_{k,j} - \eta_{k,j} \nabla \ell(z_j^{(k)}; \varphi_{k,j})$<br>    – Deploy $\theta_{k+1} = \varphi_{k,n(k)+1}$ |

Figure 1: Stochastic gradient method for performative prediction. Greedy deploy publishes the new classifier at every step while lazy deploy performs several gradient updates before releasing the new model.

## 3.1 Greedy deploy

A natural algorithm for stochastic optimization in performative prediction is a direct extension of the stochastic gradient method, whereby at every time step, we observe a sample $z^{(k)} \sim \mathcal{D}(\theta_k)$, compute a gradient update to the current model parameters $\theta_k$, and deploy the new classifier $\theta_{k+1}$ (see left panel in Figure 1). We call this algorithm *greedy deploy*. In the context of our previous traffic prediction example, this greedy procedure corresponds to iteratively updating and redeploying the model based off information from the most recent trip.

While this procedure is algorithmically identical to the stochastic gradient method in traditional convex optimization, in performative prediction, the distribution of the observed samples depends on the trajectory of the algorithm. We begin by stating a technical lemma which introduces a recursion for the distance between $\theta_k$ and $\theta_{\mathrm{PS}}$.

**Lemma 3.1.** *Assume* (A1), (A2) *and* (A3). *If the distribution map $\mathcal{D}(\cdot)$ is $\epsilon$-sensitive with $\epsilon < \gamma/\beta$, then greedy deploy with step size $\eta_k$ satisfies the following recursion for all $k \geq 1$:*

$$\mathbb{E}\left[\|\theta_{k+1} - \theta_{\mathrm{PS}}\|_2^2\right] \leq \left(1 - 2\eta_k(\gamma - \epsilon\beta) + \eta_k^2 L^2 \left(1 + \epsilon\frac{\beta}{\gamma}\right)^2\right) \mathbb{E}\left[\|\theta_k - \theta_{\mathrm{PS}}\|_2^2\right] + \eta_k^2 \sigma^2.$$

Similar recursions underlie many proofs of SGD, and Lemma 3.1 can be seen as their generalization to the performative setting. Furthermore, we see how the bound implies a strong contraction to the performatively stable point if the performative effects are weak, that is when $\epsilon \ll \gamma/\beta$.

Using this recursion, a simple induction argument suffices to prove that greedy deploy converges to the performatively stable solution (see Appendix D). Moreover, it does so at the usual $O(1/k)$ rate.

**Theorem 3.2.** *Assume* (A1), (A2) *and* (A3). *If the distribution map $\mathcal{D}(\cdot)$ is $\epsilon$-sensitive with $\epsilon < \frac{\gamma}{\beta}$, then for all $k \geq 0$ greedy deploy with step size $\eta_k = \left((\gamma - \epsilon\beta)k + 8L^2/(\gamma - \epsilon\beta)\right)^{-1}$ satisfies*

$$\mathbb{E}\left[\|\theta_{k+1} - \theta_{\mathrm{PS}}\|_2^2\right] \leq \frac{M_{\mathrm{greedy}}}{(\gamma - \epsilon\beta)^2 k + 8L^2},$$

*where $M_{\mathrm{greedy}} = \max\left\{2\sigma^2, 8L^2\|\theta_1 - \theta_{\mathrm{PS}}\|_2^2\right\}$.*

Comparing this result to the traditional analysis of SGD for smooth, strongly convex objectives (e.g. [18]), we see that the traditional factor of $\gamma$ is replaced by $\gamma - \epsilon\beta$, which we view as the effective strong convexity parameter of the performative prediction problem. When $\epsilon = 0$, there are no performative effects and the problem of finding the stable solution reduces to that of finding the risk minimizer on a fixed, static distribution. Consequently, it is natural for the two bounds to identify.

## 3.2 Lazy deploy

Contrary to greedy deploy, lazy deploy collects multiple data points and hence takes multiple stochastic gradient steps between consecutive model deployments. In the setting from Example 1,

this corresponds to observing the traffic conditions across multiple days, and potentially diverse conditions, before deploying a new model. This modification significantly changes the trajectory of lazy deploy relative to greedy deploy, given that the observed samples follow the distribution of the last *deployed* model, which might differ from the current iterate. More precisely, after deploying $\theta_k$, we perform $n(k)$ stochastic gradient steps to the model parameters, using samples from $\mathcal{D}(\theta_k)$ before we deploy the last iterate as $\theta_{k+1}$ (see right panel in Figure 1).

At a high level, lazy deploy converges to performative stability because it progressively approximates *repeated risk minimization* (RRM), defined recursively as,

$$\theta_{k+1} = \arg\min_{\theta' \in \Theta} \mathbb{E}_{z \sim \mathcal{D}(\theta_k)} \ell(z; \theta') \quad \text{for } k \geq 1 \quad \text{and } \theta_1 \in \Theta \text{ initialized arbitrarily.}$$

Perdomo et al. [17] show that RRM converges to a performatively stable classifier at a linear rate when $\epsilon < \gamma/\beta$. Since the underlying distribution remains static between deployments, a classical analysis of SGD shows that for large $n(k)$ these "offline" iterates $\varphi_{k,j}$ converge to the risk minimizer on the distribution corresponding to the previously deployed classifier. In particular, for large $n(k)$, $\theta_{k+1} \approx G(\theta_k)$. By virtue of approximately tracing out the trajectory of RRM, lazy deploy converges to $\theta_{\mathrm{PS}}$ as well. This sketch is formalized in the following theorem. For details we refer to Appendix E.

**Theorem 3.3.** *Assume (A1), (A2), and (A3), and that the distribution map $\mathcal{D}(\cdot)$ is $\epsilon$-sensitive with $\epsilon < \frac{\gamma}{\beta}$. For any $\alpha > 0$, running lazy deploy with $n(k) \geq n_0 k^\alpha$, $k = 1, 2, \ldots$ many steps between deployments and step size sequence $\eta_{k,j} = (\gamma j + 8L^2/\gamma)^{-1}$, satisfies*

$$\mathbb{E}\left[\|\theta_{k+1} - \theta_{\mathrm{PS}}\|_2^2\right] \leq c^k \cdot \|\theta_1 - \theta_{\mathrm{PS}}\|_2^2 + \left(c^{\Omega(k)} + \frac{2}{k^{\alpha \cdot (1-o(1))}}\right) \cdot M_{\mathrm{lazy}}, \tag{1}$$

*where $c = \left(\epsilon \frac{\beta}{\gamma}\right)^2 + o(1)$ and $M_{\mathrm{lazy}} = \frac{3(\sigma+\gamma)^2}{\gamma^2(1-c)}$. Here, $o(1)$ is independent of $k$ and vanishes as $n_0$ grows; $n_0$ is chosen large enough such that $c < 1$.*

### 3.3 Discussion

In this section, we have presented how varying the intervals at which we deploy models trained with stochastic gradient descent in performative prediction leads to qualitatively different algorithms. While greedy deploy resembles classical SGD with a step size sequence adapted to the strength of distribution shift, lazy deploy can be viewed as a rough approximation of repeated risk minimization.

As we alluded to previously, the convergence behavior of both algorithms is critically affected by the strength of performative effects $\epsilon$. For $\epsilon \ll \gamma/\beta$, the effective strong convexity parameter $\gamma - \epsilon\beta$ of the performative prediction problem is large. In this setting, the relevant distribution shift of deploying a new model is neglible and greedy deploy behaves almost like SGD in classical supervised learning, converging quickly to performative stability.

Conversely, for $\epsilon$ close to the convergence threshold, the contraction of greedy deploy to the performatively stable classifier is weak. In this regime, we expect lazy deploy to perform better since the convergence of the offline iterates $\varphi_{k,j}$ to the risk minimizer on the current distribution $G(\theta_k)$ is unaffected by the value of $\epsilon$. Lazy deploy then converges by closely mimicking the behavior of RRM.

Furthermore, both algorithms differ in their sensitivity to different initializations. In greedy deploy, the initial distance $\|\theta_1 - \theta_{\mathrm{PS}}\|_2^2$ decays polynomially, while in lazy deploy it decays at a linear rate. This suggests that the lazy deploy algorithm is more robust to poor initialization. While we derive these insights purely by inspecting our upper bounds, we find that these heuristic observations also hold empirically, as shown in the next section.

In terms of the asymptotics of both algorithms, we identify the following tradeoff between the number of samples and the number of deployments sufficient to converge to performative stability.

**Corollary 3.4.** *Assume (A1), (A2), and (A3), and that $\mathcal{D}(\cdot)$ is $\epsilon$-sensitive with $\epsilon < \frac{\gamma}{\beta}$.*

- *To ensure that greedy deploy returns a solution $\theta^\star$ such that, $\mathbb{E}\left[\|\theta^\star - \theta_{\mathrm{PS}}\|_2^2\right] \leq \delta$, it suffices to collect $\mathcal{O}(1 / \delta)$ samples and to deploy $\mathcal{O}(1 / \delta)$ classifiers.*
- *To achieve the same guarantee using lazy deploy, it suffices to collect $\mathcal{O}(1 / \delta^{\frac{\alpha+1}{(1-\omega)\cdot\alpha}})$ samples and to deploy $\mathcal{O}(1 / \delta^{\frac{1}{\alpha}})$ classifiers, for any $\alpha > 0$ and some $\omega = 1 - o(1)$ which tends to 1 as $n_0$ grows.*

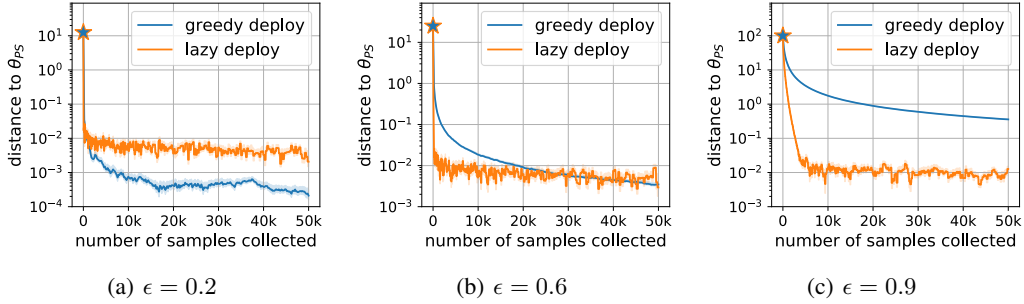

(a) $\epsilon = 0.2$        (b) $\epsilon = 0.6$        (c) $\epsilon = 0.9$

Figure 2: Convergence of lazy and greedy deploy to performative stability for varying values of $\epsilon$. We use $n(k) = k$ for lazy deploy. The results are for the synthetic Gaussian example with $\mu = 10$, $\sigma = 0.1$.

We see from the above result that by choosing large enough values of $n_0$ and $\alpha$, we can make the sample complexity of the lazy deploy algorithm come arbitrarily close to that of greedy deploy. However, to match the same convergence guarantee, lazy deploy only performs $\mathcal{O}(1 \,/\, \delta^{\frac{1}{\alpha}})$ deployments, which is significantly better than the $\mathcal{O}(1 \,/\, \delta)$ deployments for greedy.

This reduction in the number of deployments is particularly relevant when considering the settings that performative prediction is meant to address. Whenever we use prediction in social settings, there are important social costs associated with making users adapt to a new model [15]. Furthermore, in industry, there are often significant technical challenges associated with deploying a new classifier [2]. By choosing $n(k) = n_0 k^\alpha$ appropriately, we can reduce the number of deployments necessary for lazy deploy to converge while at the same time improving the sample complexity of the algorithm.

## 4 Experiments

We complement our theoretical analysis of greedy and lazy deploy with a series of empirical evaluations. First, we carry out experiments using synthetic data where we can analytically compute stable points and carefully evaluate the tradeoffs suggested by our theory. Second, we evaluate the performance of these procedures on a strategic classification simulator previously used as a benchmark for optimization in performative settings by [17].

### 4.1 Synthetic data

For our first experiment, we consider the task of estimating the mean of a Gaussian random variable under performative effects. In particular, we consider minimizing the expected squared loss $\ell(z; \theta) = \frac{1}{2}(z - \theta)^2$ where $z \sim \mathcal{D}(\theta) = \mathcal{N}(\mu + \epsilon\theta, \sigma^2)$. For $\epsilon > 0$, the true mean of a distribution $\mathcal{D}(\theta)$ depends on our revealed estimate $\theta$. Furthermore, for $\epsilon < \gamma/\beta = 1$, the problem has a unique stable point. A short algebraic manipulation shows that $\theta_{\mathrm{PS}} = \frac{\mu}{1-\epsilon}$. As per our theory, both greedy and lazy deploy converge to performative stability for all $\epsilon < 1$.

**Effect of performativity.** We compare the convergence behavior of lazy deploy and greedy deploy for various values of $\epsilon$ in Figure 2. We choose step sizes for both algorithms according to our theorems in Section 3. In the case of lazy deploy, we set $\alpha = 1$, and hence $n(k) \propto k$.

We see that when performative effects are weak, i.e. $\epsilon \ll \gamma/\beta$, greedy deploy outperforms lazy. Lazy deploy in turn is better at coping with large distribution shifts from strong performative effects. These results confirm the conclusions from our theory and show that the choice of greedy vs lazy deployment can indeed have a large impact on algorithm performance depending on the value of $\epsilon$.

**Deployment schedules.** We also experiment with different deployment schedules $n(k)$ for lazy deploy. As described in Theorem 3.3, we can choose $n(k) \propto k^\alpha$ for all $\alpha > 0$. The results for $\alpha \in \{0.5, 1, 2\}$ and $\epsilon \in \{.2, .6, .9\}$, are depicted in Figure 4 in the Appendix. We find that shorter deployment schedules, i.e., smaller $\alpha$, lead to faster progress during initial stages of the optimization, whereas longer deployments schedules fare better in the long run while at the same time significantly reducing the number of deployments.

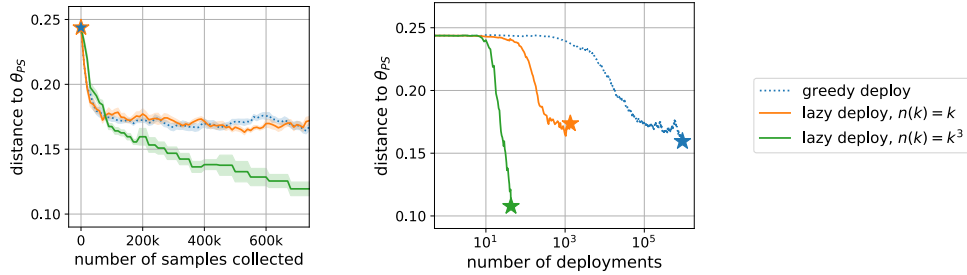

Figure 3: Convergence of lazy and greedy deploy to performative stability. Results are for the strategic classification experiments with $\epsilon = 100$. (left panel) convergence as a function of the number of samples. (right panel) convergence as a function of the number of deployments.

## 4.2 Strategic classification

In addition to the experiments on synthetic data, we also evaluate the performance of the two optimization procedures in a simulated strategic classification setting. Strategic classification is a two-player game between an institution which deploys a classifier $f_\theta$ and individual agents who manipulate their features in order to achieve a more favorable classification.

Perdomo et al. [17] introduce a credit scoring simulator in which a bank deploys a logistic regression classifier to determine the probability that an individual will default on a loan. Individuals correspond to feature, label pairs $(x, y)$ drawn from a Kaggle credit scoring dataset [9]. Given the bank's choice of a classifier $f_\theta$, individuals solve an optimization problem to compute the best-response set of features, $x_{\mathrm{BR}}$. This optimization procedure is parameterized by a value $\epsilon$ which determines the extent to which agents can change their respective features. The bank then observes the manipulated data points $(x_{\mathrm{BR}}, y)$. This data-generating process can be described by a distribution map, which we can verify is $\epsilon$-sensitive. For additional details we refer to Appendix A.

At each time step, the learner observes a single sample from the distribution in which the individual's features have been manipulated in response to the most recently deployed classifier. This is in contrast to the experimental setup in [17], where the learner gets to observe the entire distribution of manipulated features at every step. While we cannot compute the stable point analytically in this setting, we can calculate it empirically by running RRM until convergence.

**Results.** The inverse condition number of this problem is much smaller than in the Gaussian example; we have $\gamma/\beta \approx 10^{-2}$. We fist pick $\epsilon$ within the regime of provable convergence, i.e., $\epsilon = 10^{-3}$, and compare the two methods. As expected, for such a small value of $\epsilon$ greedy deploy is the preferred method. Results are depicted in Figure 6 in the Appendix.

Furthermore, we explore the behavior of these algorithms outside the regime of provable convergence with $\epsilon \gg \gamma/\beta$. We choose step sizes for both algorithms as defined in Section 3 with the exception that we ignore the $\epsilon$-dependence in the step size schedule of greedy deploy and choose the same initial step size as for lazy deploy (Theorem 3.2). As illustrated in Figure 3 (left), lazy significantly outperforms greedy deploy in this setting. Moreover, the performance of lazy deploy significantly improves with $\alpha$. In addition to speeding up convergence, choosing larger sample collection schedules $n(k)$ substantially reduces the number of deployments, as seen in Figure 3 (right).

## Funding disclosure

This research was supported by an NSF CAREER award (#1750555). MH is a paid consultant for Twitter. He was previously a paid consultant for Google. CMD is supported by the Swiss National Science Foundation Postdoc Mobility Fellowship Program. JCP is in part supported by the U.S. National Science Foundation Graduate Research Fellowship Program.

## Broader impact statement

The motivation for studying performative prediction comes from the observation that whenever we use supervised learning in social settings, we almost never make predictions for predictions' sake, but rather to inform decision making within some broader context [11]. Banks predict default risks to decide to whom they will allocate loans. Commuters use estimated time of arrival (ETA) prediction to choose which route to take to work. Governments predict crime rates to decide how to deploy police forces [8, 13]. In each of these settings, our choice of predictive model leads to changes in the way the broader system behaves and hence in the distribution over observed data.

Our work introduces optimization procedures for finding classifiers with good predictive performance for these performative settings. Here, we use good to indicate that these models are accurate. However, as is clear from examples in recent history, the societal impacts of having an accurate model depend on the context in which prediction is used, and the intent of the system designer. As a society, we can benefit from having robust and reliable systems to forecast congestion in cities, yet it would be remiss of us to overlook how these advances could also be used to infringe upon civil liberties.

As a subfield of learning theory, performative prediction is only just starting to receive attention from the community and papers in this area are largely theoretical in nature. Therefore, much remains to be seen in terms of the broader impact of these ideas. We eagerly welcome feedback and comments from members of the community as to how we may ensure the success of this research agenda.

## Footnotes

†MH is a paid consultant for Twitter.

[3]Gradients of the loss $\ell$ are always taken with respect to the parameters $\theta$.

[4]The reason is that the notion of *joint* smoothness we consider does not scale like strong convexity when rescaling $\theta$. For example, rescaling $\theta \mapsto 2\theta$ (thus making $\epsilon \mapsto \epsilon/2$) would downscale the strong convexity parameter and the parameter corresponding to the usual notion of smoothness in optimization by 4, however the smoothness in $z$ would downscale by 2.

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
