[Supplementary Material]

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

 collected for various values of $\epsilon$. (bottom row) We plot convergence in the same setting, but now as a function of the number of deployments. For comparison we add greedy deploy (red) and RRM (dashed, gray line). The stars indicate the value attained at the end of our simulation (50k SGD updates).

## A  Additional evaluations and details on experimental setup

**Algorithm parameters.** If not stated otherwise we use the step size schedule proposed by our theory:

- greedy deploy (Theorem 3.3): $\eta_{k,j} = \frac{c_\eta}{j+k_0}$, where $c_\eta = \frac{1}{\gamma}$ and $k_0 = \frac{8L^2}{\gamma^2}$.
- lazy deploy (Theorem 3.2): $\eta_k = \frac{c_\eta}{k+k_0}$, where $c_\eta = \frac{1}{\gamma-\epsilon\beta}$ and $k_0 = \frac{8L^2}{(\gamma-\epsilon\beta)^2}$.

In Figure 3, since we experiment with $\epsilon = 100$ which is outside the regime of our theory, we adapt the $\epsilon$-dependence of the step size in greedy deploy. In particular, we pick $c_\eta = \frac{100}{\gamma}$ and $k_0 = \frac{8L^2}{\gamma^2}$ for both algorithms. The factor 100 was found empirically to reduce runtime.

The deployment schedule $n(k)$ for lazy deploy is parameterized by $\alpha$ as $n(k) = n_0 k^\alpha$, where we choose $n_0 = 1$ for our experiments.

**Confidence invervals.** We repeat all our experiments 30 times and plot the mean $\mu_s$ and the shaded area $\mu_s \pm z \frac{s}{\sqrt{n}}$ where $s$ denotes the standard deviation computed over the runs and $z = 1.645$. The value of $z$ is chosen to ensure 90% coverage assuming Gaussian errors in the data.

### A.1  Details for synthetic Gaussian experiments

The distribution map for the synthetic example is given by $\mathcal{D}(\theta) = \mathcal{N}(\mu + \theta\epsilon, \sigma^2)$ where we use $\mu = 10$ and $\sigma = 0.1$ for our experiments. The SGD updates take the following form:

- greedy deploy: $\theta_{k+1} = \theta_k + \eta_k(z^{(k)} - \theta_k)$ where $z^{(k)} \sim \mathcal{D}(\theta_k)$.

– lazy deploy: $\varphi_{k,j+1} = \varphi_{k,j} + \eta_{k,j}(z_j^{(k)} - \varphi_{k,j})$, where $z_j^{(k)} \sim \mathcal{D}(\theta_k)$.

We initialize all optimization procedures at the risk minimizer $\theta_1 = \mu$ to mitigate effects of bad initialization and to instead focus on the effects of performativity.

## A.2 Strategic classification simulator

For these experiments, we use the same experimental setup used by Perdomo et al. [17] as implemented in the WhyNot library [14]. We nevertheless include the all the relevant details for the sake of completeness.

---

**Input:** base distribution $\mathcal{D}$, classifier $f_\theta$, cost function $c$, and utility function $u$
**Sampling procedure for** $\mathcal{D}(\theta)$**:**
    1. Sample $(x, y) \sim \mathcal{D}$
    2. Compute best response $x_{\mathrm{BR}} \leftarrow \arg\max_{x'} u(x', \theta) - c(x', x)$
    3. Output sample $(x_{\mathrm{BR}},\ y)$

---

Figure 5: Distribution map for strategic classification (Perdomo et al. [17]).

The distribution map for this strategic classification example is described in Figure 5. The base distribution $\mathcal{D}$ is a subsampled version of the Kaggle dataset [9] with $d = 10$ features and $n = 18357$ examples. Labels are binary variables $y \in \{0, 1\}$ and indicate whether an individual defaulted on a loan or not. We preprocess the data and normalize features to have zero mean and unit standard deviation. Out of the ten features, three are treated as strategic features. These are dimensions $1, 6, 8$ corresponding to features such as the number of open credit lines.

The empirical distribution on these 18k points is considered to be the true distribution. To run our stochastic optimization experiments, we simply sample a single example from the data set according to the data-generating process described in Figure 5.

Individual utilities $u(\theta, x) = -\theta^\top x$ are linear and the costs are quadratic $c(x', x) = \frac{1}{2\epsilon}\|x' - x\|$. Together, these lead to an $\epsilon$-sensitive distribution map as shown in [17].

The loss of the institution is a logistic loss with $\ell_2$ regularization:

$$\frac{1}{n} \sum_{i=1}^{n} \left[ \log(1 + \exp(x_i^\top \theta)) - y_i x_i^\top \theta \right] + \frac{\lambda}{2}\|\theta\|^2$$

This loss is $\gamma$-strongly convex and $\beta = \max\left\{2, \frac{1}{4n}\sum_{i=1}^{n}\|x_i\|_2^2 + \gamma\right\}$. jointly smooth [17]. We fix $\lambda = 10^3/n$ for all experiments. When evaluated on the base distribution, the objective has parameters $\beta = 4.72, \gamma = 0.054$ which yields $\frac{\gamma}{\beta} = 0.011$.

Figure 6: Convergence of lazy and greedy deploy to performative stability. Results are for the strategic classification experiments with $\epsilon = 0.001$. (left panel) convergence as a function of the number of samples collected. (center panel) convergence as a function of the number of deployments. (right panel) excess performative risk with respect the the stable classifier $\theta_{\mathrm{PS}}$ as a function of stochastic gradient updates.

# B Technical lemmas

**Lemma B.1** (Kantorovich-Rubinstein). *A distribution map $\mathcal{D}(\cdot)$ is $\epsilon$-sensitive if and only if for all $\theta, \theta' \in \Theta$:*

$$\sup\left\{ \left| \underset{Z\sim\mathcal{D}(\theta)}{\mathbb{E}} g(Z) - \underset{Z\sim\mathcal{D}(\theta')}{\mathbb{E}} g(Z) \right| \leq \epsilon\|\theta - \theta'\|_2 \ : \ g : \mathbb{R}^p \to \mathbb{R}, \ g \text{ 1-Lipschitz}\right\}.$$

**Lemma B.2** (Lemma C.4 in [17]). *Let $f : \mathbb{R}^n \to \mathbb{R}^d$ be an L-Lipschitz function, and let $X, X' \in \mathbb{R}^n$ be random variables such that $W_1(X, X') \leq C$. Then*

$$\| \mathbb{E}[f(X)] - \mathbb{E}[f(X')]\|_2 \leq LC.$$

**Lemma B.3** (First-order optimality condition). *Let $f$ be convex and let $\Omega$ be a closed convex set on which $f$ is differentiable, then*

$$x_* \in \underset{x\in\Omega}{\arg\min} f(x)$$

*if and only if*

$$\nabla f(x_*)^T(y - x_*) \geq 0, \quad \forall y \in \Omega.$$

**Lemma B.4** (Theorem 3.5 in [17]). *Suppose the loss function is $\gamma$-strongly convex (A2) and $\beta$-jointly smooth (A3). Then, for all $\theta, \theta' \in \Theta$, it holds that,*

$$\|G(\theta) - G(\theta')\|_2 \leq \epsilon\frac{\gamma}{\beta}\|\theta - \theta'\|_2.$$

**Lemma B.5.** *Let $s \in (0,1)$, and fix $\alpha > 0$, then,*

$$\sum_{k=1}^{t} k^{-\alpha} s^{t-k} \leq \frac{s^{t(1-2^{-1/\alpha})}}{1-s} + \frac{2t^{-\alpha}}{1-s}.$$

*Proof.* Denote by $a_k \overset{\text{def}}{=} k^{-\alpha}$. Let $M_t = \max\{m \in \mathbb{N} : a_m > 2a_t\}$. We decompose the sum depending on $M_t$ as follows:

$$\sum_{k=1}^{t} a_k s^{t-k} = \sum_{k=1}^{M_t} a_k s^{t-k} + \sum_{k=M_t+1}^{t} a_k s^{t-k}.$$

We bound the first term trivially, by applying the fact that $a_k \leq 1$. For the second term, we use the fact that $a_k \leq 2a_t$ for $k > M_t$. We thus get:

$$\sum_{k=1}^{t} a_k s^{t-k} \leq \sum_{k=1}^{M_t} s^{t-k} + 2a_t \sum_{k=M_t+1}^{t} s^{t-k} \leq \frac{s^{t-M_t}}{1-s} + \frac{2a_t}{1-s}.$$

Since $a_k = k^{-\alpha}$, then $M_t \leq \frac{t}{2^{1/\alpha}}$, and so

$$\frac{s^{t-M_t}}{1-s} + \frac{2a_t}{1-s} \leq \frac{s^{t(1-2^{-1/\alpha})}}{1-s} + \frac{2a_t}{1-s}.$$

$\square$

# C Population-level results: proofs

## C.1 Proof of Proposition 2.2

First we state a counterexample to prove claim (a), and then we generalize our construction to prove claim (b).

**a)** Let $\Theta = \mathbb{R}^2$, and let $z \sim \mathcal{D}(\theta)$ be a point mass at $\epsilon\theta$. This distribution map is clearly $\epsilon$-sensitive. Furthermore, define the loss as,

$$\ell(z;\theta) = \beta \cdot \theta^\top \begin{bmatrix} 0 & -1 \\ 1 & 0 \end{bmatrix} z,$$

where $\beta$ is an arbitrary positive scalar. Note that this objective is linear in $\theta$ and hence convex. Furthermore, it is $\beta$-jointly smooth and has a unique performatively stable point at the origin.

Repeated gradient descent has the dynamics:

$$\theta_{k+1} = \theta_k - \eta_k \mathop{\mathbb{E}}_{z \sim \mathcal{D}(\theta_k)} \nabla\ell(z;\theta_k)$$

$$= \theta_k - \eta_k \epsilon\beta \begin{bmatrix} 0 & -1 \\ 1 & 0 \end{bmatrix} \theta_k.$$

In particular, we can write

$$\theta_{t+1} = \begin{bmatrix} 1 & c_k \\ -c_k & 1 \end{bmatrix} \theta_k \stackrel{\text{def}}{=} C_k\theta_k,$$

where $c_k = \eta_k\beta\epsilon > 0$. The matrix $C_k$ has eigenvalues $1 \pm ic$ and hence $|\lambda_1| = |\lambda_2| = \sqrt{1+c^2} > 1$. Therefore, $\|\theta_{t+1}\|_2 > \|\theta_t\|_2$, thus proving that repeated gradient descent cannot converge.

Intuitively, if we initialize RGD at any point other than the stable point at origin in this example, gradient flow "spirals outwards".

**b)** Now we generalize claim (a) to show that $\epsilon < \frac{\gamma}{\beta}$ is a sharp threshold for convergence of RGD. Let $\Theta = \mathbb{R}^2$ and let $z \sim \mathcal{D}(\theta)$ be a point mass at $\epsilon\theta$. As before, this map is clearly $\epsilon$-sensitive. Define the loss to be

$$\ell(z;\theta) = \theta^\top \begin{bmatrix} a & -b \\ b & a \end{bmatrix} z + \frac{\gamma}{2}\|\theta\|_2^2.$$

This loss is $\gamma$-strongly convex in $\theta$. Notice that the gradient

$$\nabla\ell(z;\theta) = \begin{bmatrix} a & -b \\ b & a \end{bmatrix} z + \gamma\theta$$

is $\beta = \max\{\gamma, a^2 + b^2\}$-jointly smooth, given that it is $\gamma$-Lipschitz in $\theta$ and $a^2 + b^2$-Lipschitz in $z$. Here, $a^2 + b^2$ is the operator norm of the 2x2 matrix. Repeated gradient descent performs the following update:

$$\theta_{k+1} = \theta_k - \eta_k \mathop{\mathbb{E}}_{z \sim \mathcal{D}(\theta_k)} \nabla\ell(z;\theta_k)$$

$$= \begin{bmatrix} 1 - \eta_k(\epsilon a + \gamma) & \eta_k\epsilon b \\ -\eta_k\epsilon b & 1 - \eta_k(\epsilon a + \gamma) \end{bmatrix} \theta_k.$$

If we set $b = \sqrt{\beta - a^2}$ for $\beta \geq \gamma$, then the loss is $\beta$-jointly smooth. Assume that $\beta < 1$, and set $a = -\beta$. If $\epsilon \geq \frac{\gamma}{\beta}$, then the update becomes of the form:

$$\theta_{k+1} = \begin{bmatrix} 1 + c_{k,0} & c_{k,1} \\ -c_{k,1} & 1 + c_{k,0} \end{bmatrix} \theta_k \stackrel{\text{def}}{=} C_k\theta_k,$$

for some $c_{k,0} \geq 0, c_{k,1} > 0$. The matrix $C_k$ has eigenvalues equal to $1 + c_{k,0} \pm ic_{k,1}$, and these eigenvalues have modulus equal to $(1 + c_{k,0})^2 + c_{k,1}^2 > 1$. Therefore, RGD is an expansive update for all choices of step size $\eta_k > 0$, implying that it cannot converge.

## C.2 Proof of Proposition 2.3

This proof is essentially a consequence of Lemma 3.1. By following the steps of Lemma 3.1, we get

$$\|\theta_{k+1} - \theta_{\text{PS}}\|_2^2 \leq \|\theta_k - \theta_{\text{PS}}\|_2^2 - 2\eta_k(\mathbb{E}\,\nabla\ell(z^{(k)};\theta_k))^\top(\theta_k - \theta_{\text{PS}}) + \eta^2 \|\mathbb{E}\,\nabla\ell(z^{(k)};\theta_k)\|_2^2$$

$$\stackrel{\text{def}}{=} B_1 - 2\eta B_2 + \eta^2 B_3.$$

Following the same approach as in Lemma 3.1, we get
$$B_2 \geq (\gamma - \epsilon\beta)\|\theta_k - \theta_{\mathrm{PS}}\|_2^2.$$
The bound on $B_3$ is slightly different, as we no longer make assumptions on the second moment of the gradients; we proceed as follows:

$$
\begin{aligned}
\|\mathbb{E}\,\nabla\ell(z^{(k)};\theta_k)\|_2^2 &= \|\mathbb{E}\,\nabla\ell(z^{(k)};\theta_k) - \mathbb{E}\,\nabla\ell(z^{(\theta_{\mathrm{PS}})};\theta_{\mathrm{PS}})\|_2^2 \\
&\leq \|\mathbb{E}\,\nabla\ell(z^{(k)};\theta_k) - \mathbb{E}\,\nabla\ell(z^{(k)};\theta_{\mathrm{PS}}) + \mathbb{E}\,\nabla\ell(z^{(k)};\theta_{\mathrm{PS}}) - \mathbb{E}\,\nabla\ell(z^{(\theta_{\mathrm{PS}})};\theta_{\mathrm{PS}})\|_2^2 \\
&\leq 2\|\mathbb{E}\,\nabla\ell(z^{(k)};\theta_k) - \mathbb{E}\,\nabla\ell(z^{(k)};\theta_{\mathrm{PS}})\|_2^2 \\
&\quad + 2\|\mathbb{E}\,\nabla\ell(z^{(k)};\theta_{\mathrm{PS}}) - \mathbb{E}\,\nabla\ell(z^{(\theta_{\mathrm{PS}})};\theta_{\mathrm{PS}})\|_2^2 \\
&\leq 2\beta^2\|\theta_k - \theta_{\mathrm{PS}}\|_2^2 + 2\beta^2\epsilon^2\|\theta_k - \theta_{\mathrm{PS}}\|_2^2 \\
&\leq 2\beta^2\left(1 + \epsilon^2\right)\|\theta_k - \theta_{\mathrm{PS}}\|_2^2,
\end{aligned}
$$

where in the third inequality we apply the fact that the loss if $\beta$-jointly smooth, together with Lemma B.2. Putting everything together, this implies
$$\|\theta_{k+1} - \theta_{\mathrm{PS}}\|_2^2 \leq (1 - 2\eta(\gamma - \epsilon\beta) + 2\eta^2\beta^2(1+\epsilon^2))\|\theta_k - \theta_{\mathrm{PS}}\|_2^2.$$
Using the fact that $\sqrt{1-x} \leq 1 - \frac{x}{2}$ for $x \in [0,1]$, we get
$$\|\theta_{k+1} - \theta_{\mathrm{PS}}\|_2 \leq (1 - \eta(\gamma - \epsilon\beta) + \eta^2\beta^2(1+\epsilon^2))\|\theta_k - \theta_{\mathrm{PS}}\|_2.$$
By setting $\eta = \frac{\gamma - \epsilon\beta}{2(1+\epsilon^2)\beta^2}$, we can conclude

$$\|\theta_{k+1} - \theta_{\mathrm{PS}}\|_2 \leq \left(1 - \frac{(\gamma - \epsilon\beta)^2}{4(1+\epsilon^2)\beta^2}\right)\|\theta_k - \theta_{\mathrm{PS}}\|_2.$$

Note that $\frac{(\gamma-\epsilon\beta)^2}{4(1+\epsilon^2)\beta^2} < 1$ because $(\gamma - \epsilon\beta)^2 \leq \gamma^2 + \epsilon^2\beta^2 \leq (1+\epsilon^2)\beta^2$.

We can unroll the above recursion to get

$$
\begin{aligned}
\|\theta_{k+1} - \theta_{\mathrm{PS}}\|_2 &\leq \left(1 - \frac{(\gamma - \epsilon\beta)^2}{4(1+\epsilon^2)\beta^2}\right)^k \|\theta_1 - \theta_{\mathrm{PS}}\|_2 \\
&\leq \exp\left(-\frac{k(\gamma - \epsilon\beta)^2}{4(1+\epsilon^2)\beta^2}\right)\|\theta_1 - \theta_{\mathrm{PS}}\|_2.
\end{aligned}
$$

Setting the right-hand side to $\delta$ and expressing $k$ completes the proof.

## D  Greedy deploy: proofs

### D.1  Proof of Lemma 3.1

Throughout the proof, we will use $z^{(\theta_{\mathrm{PS}})}$ to denote a sample from $\mathcal{D}(\theta_{\mathrm{PS}})$ which is independent from the whole trajectory of greedy deploy (e.g. $\{\theta_j, z^{(j)}\}_j$, etc.).

Since $\Theta$ is closed and convex, we know
$$\|\theta_{k+1} - \theta_{\mathrm{PS}}\|_2^2 = \|\Pi_\Theta(\theta_k - \eta_k\nabla\ell(z^{(k)};\theta_k)) - \theta_{\mathrm{PS}}\|_2^2 \leq \|\theta_k - \eta_k\nabla\ell(z^{(k)};\theta_k) - \theta_{\mathrm{PS}}\|_2^2.$$
Squaring the right-hand side and expanding out the square,

$$
\begin{aligned}
&\mathbb{E}\left[\|\theta_k - \eta_k\nabla\ell(z^{(k)};\theta_k) - \theta_{\mathrm{PS}}\|_2^2\right] \\
&= \mathbb{E}\left[\|\theta_k - \theta_{\mathrm{PS}}\|_2^2\right] - 2\eta_k\mathbb{E}\left[\nabla\ell(z^{(k)};\theta_k)^\top(\theta_k - \theta_{\mathrm{PS}})\right] + \eta_k^2\mathbb{E}\left[\|\nabla\ell(z^{(k)};\theta_k)\|_2^2\right] \\
&\stackrel{\text{def}}{=} B_1 - 2\eta_k B_2 + \eta_k^2 B_3.
\end{aligned}
$$

We begin by lower bounding $B_2$. Since $\theta_{\mathrm{PS}}$ is optimal for the distribution it induces, by Lemma B.3 we have $\mathbb{E}\left[\nabla\ell(z^{(\theta_{\mathrm{PS}})};\theta_{\mathrm{PS}})^\top(\theta_k - \theta_{\mathrm{PS}})\right] \geq 0$. This allows us to bound $B_2$ as:

$$
\begin{aligned}
B_2 &\geq \mathbb{E}\left[(\nabla\ell(z^{(k)};\theta_k) - \nabla\ell(z^{(\theta_{\mathrm{PS}})};\theta_k) + \nabla\ell(z^{(\theta_{\mathrm{PS}})};\theta_k) - \nabla\ell(z^{(\theta_{\mathrm{PS}})};\theta_{\mathrm{PS}}))^\top(\theta_k - \theta_{\mathrm{PS}})\right] \\
&= \mathbb{E}\left[(\nabla\ell(z^{(k)};\theta_k) - \nabla\ell(z^{(\theta_{\mathrm{PS}})};\theta_k)^\top(\theta_k - \theta_{\mathrm{PS}})\right] \\
&\quad + \mathbb{E}\left[(\nabla\ell(z^{(\theta_{\mathrm{PS}})};\theta_k) - \nabla\ell(z^{(\theta_{\mathrm{PS}})};\theta_{\mathrm{PS}}))^\top(\theta_k - \theta_{\mathrm{PS}})\right].
\end{aligned}
$$

For the first term, we have that

$$\mathbb{E}\left[\left(\nabla\ell(z^{(k)};\theta_k) - \nabla\ell(z^{(\theta_{\mathrm{PS}})};\theta_k)\right)^\top(\theta_k - \theta_{\mathrm{PS}})\right]$$

$$= \mathbb{E}\left[\mathbb{E}\left[\left(\nabla\ell(z^{(k)};\theta_k) - \nabla\ell(z^{(\theta_{\mathrm{PS}})};\theta_k)^\top(\theta_k - \theta_{\mathrm{PS}}) \mid \theta_k\right]\right]$$

$$\geq -\epsilon\beta\,\mathbb{E}\left[\|\theta_k - \theta_{\mathrm{PS}}\|_2^2\right].$$

Having applied the law of iterated expectation, the above inequality follows from the fact that, conditional on $\theta_k$, the function $\nabla\ell(z;\theta_k)^\top(\theta_k - \theta_{\mathrm{PS}})$ is $\beta\|\theta_k - \theta_{\mathrm{PS}}\|_2$–Lipschitz in $z$. To verify this claim, we can apply the Cauchy-Schwarz inequality followed by the fact that the gradient is $\beta$-jointly smooth. Then, we apply Lemma B.1 and the fact that $\mathcal{D}(\cdot)$ is $\epsilon$-sensitive to get the final bound.

Now, we use strong convexity to bound the second term,

$$\mathbb{E}\left[\left(\nabla\ell(z^{(\theta_{\mathrm{PS}})};\theta_k) - \nabla\ell(z^{(\theta_{\mathrm{PS}})};\theta_{\mathrm{PS}})\right)^\top(\theta_k - \theta_{\mathrm{PS}})\right]$$

$$= \mathbb{E}\left[\mathbb{E}\left[\left(\nabla\ell(z^{(\theta_{\mathrm{PS}})};\theta_k) - \nabla\ell(z^{(\theta_{\mathrm{PS}})};\theta_{\mathrm{PS}})\right)^\top(\theta_k - \theta_{\mathrm{PS}}) \mid \theta_k\right]\right]$$

$$\geq \gamma\,\mathbb{E}\left[\|\theta_k - \theta_{\mathrm{PS}}\|_2^2\right].$$

Therefore, we get that

$$B_2 \geq (\gamma - \epsilon\beta)\,\mathbb{E}\left[\|\theta_k - \theta_{\mathrm{PS}}\|_2^2\right].$$

Now we move on to bounding $B_3$. Using our assumption on the variance on the gradients yields the following bound, we get

$$\mathbb{E}\left[\|\nabla\ell(z^{(k)};\theta_k)\|_2^2\right] \leq \sigma^2 + L^2\,\mathbb{E}\left[\|\theta_k - G(\theta_k)\|_2^2\right]$$

$$= \sigma^2 + L^2\,\mathbb{E}\left[\|\theta_k - \theta_{\mathrm{PS}} + \theta_{\mathrm{PS}} - G(\theta_k)\|_2^2\right]$$

$$\leq \sigma^2 + L^2\left(\mathbb{E}\left[\left(\|\theta_k - \theta_{\mathrm{PS}}\|_2 + \|\theta_{\mathrm{PS}} - G(\theta_k)\|_2\right)^2\right]\right)$$

$$\leq \sigma^2 + L^2\left(1 + \epsilon\frac{\beta}{\gamma}\right)^2\mathbb{E}\left[\|\theta_k - \theta_{\mathrm{PS}}\|_2^2\right],$$

where in the last step we use Lemma B.4, which implies $\|\theta_{\mathrm{PS}} - G(\theta_k)\|_2 \leq \epsilon\frac{\beta}{\gamma}\|\theta_k - \theta_{\mathrm{PS}}\|_2$.

Putting all the steps together completes the proof.

## D.2 Proof of Theorem 3.2

From Lemma 3.1, we have that the following recursion holds:

$$\mathbb{E}\left[\|\theta_{k+1} - \theta_{\mathrm{PS}}\|_2^2\right] \leq \left(1 - 2\eta_k(\gamma - \epsilon\beta) + \eta_k^2 L^2\left(1 + \epsilon\frac{\beta}{\gamma}\right)^2\right)\mathbb{E}\left[\|\theta_k - \theta_{\mathrm{PS}}\|_2^2\right] + \eta_k^2\sigma^2.$$

Using the fact that $\epsilon < \frac{\gamma}{\beta}$, we get that,

$$\mathbb{E}\left[\|\theta_{k+1} - \theta_{\mathrm{PS}}\|_2^2\right] \leq \left(1 - 2\eta_k(\gamma - \epsilon\beta) + 4\eta_k^2 L^2\right)\mathbb{E}\left[\|\theta_k - \theta_{\mathrm{PS}}\|_2^2\right] + \eta_k^2\sigma^2.$$

We proceed by using induction. As in the theorem statement, we let $\eta_k = \frac{1}{(\gamma - \epsilon\beta)(k + k_0)}$, where we denote $k_0 = \frac{8L^2}{(\gamma - \epsilon\beta)^2}$. The base case, $k = 0$, is trivially true by construction of the bound and choice of $k_0$. Now, we adopt the inductive hypothesis that

$$\mathbb{E}\left[\|\theta_{k+1} - \theta_{\mathrm{PS}}\|_2^2\right] \leq \frac{\max\left\{2\sigma^2, 8L^2\|\theta_1 - \theta_{\mathrm{PS}}\|_2^2\right\}}{(\gamma - \epsilon\beta)^2(k + k_0)}.$$

Then, by Lemma 3.1, it is true that

$$\mathbb{E}\left[\|\theta_{k+2} - \theta_{\mathrm{PS}}\|_2^2\right] \leq \left(1 - 2\eta_k(\gamma - \epsilon\beta) + 4\eta_k^2 L^2\right)\mathbb{E}\left[\|\theta_{k+1} - \theta_{\mathrm{PS}}\|_2^2\right] + \eta_k^2\sigma^2$$

$$\leq \frac{1}{(\gamma - \epsilon\beta)^2}\left(\frac{k + k_0 - 2 + \frac{4L^2}{(\gamma - \epsilon\beta)^2 k_0}}{(k + k_0)^2}\max\left\{2\sigma^2, 8L^2\|\theta_1 - \theta_{\mathrm{PS}}\|_2^2\right\} + \frac{\sigma^2}{(k + k_0)^2}\right)$$

$$\leq \frac{1}{(\gamma - \epsilon\beta)^2}\left(\frac{k + k_0 - 1.5}{(k + k_0)^2}\max\left\{2\sigma^2, 8L^2\|\theta_1 - \theta_{\mathrm{PS}}\|_2^2\right\} + \frac{\sigma^2}{(k + k_0)^2}\right)$$

$$\leq \frac{1}{(\gamma - \epsilon\beta)^2}\left(\frac{k + k_0 - 1}{(k + k_0)^2}\max\left\{2\sigma^2, 8L^2\|\theta_1 - \theta_{\mathrm{PS}}\|_2^2\right\} - \frac{0.5 \cdot 2\sigma^2 - \sigma^2}{(k + k_0)^2}\right)$$

$$= \frac{1}{(\gamma - \epsilon\beta)^2} \cdot \frac{k + k_0 - 1}{(k + k_0)^2}\max\left\{2\sigma^2, 8L^2\|\theta_1 - \theta_{\mathrm{PS}}\|_2^2\right\}$$

$$\leq \frac{1}{(\gamma - \epsilon\beta)^2} \cdot \frac{1}{k + 1 + k_0}\max\left\{2\sigma^2, 8L^2\|\theta_1 - \theta_{\mathrm{PS}}\|_2^2\right\},$$

where the last step follows because $(k + k_0)^2 > (k + k_0)^2 - 1 = (k + k_0 + 1)(k + k_0 - 1)$. Therefore, we have shown $\mathbb{E}\left[\|\theta_{k+2} - \theta_{\mathrm{PS}}\|_2^2\right] \leq \frac{M_{\mathrm{greedy}}}{(\gamma - \epsilon\beta)^2(k + 1 + k_0)}$, which completes the proof by induction.

## E Lazy deploy: proofs

To prove Theorem 3.3, we use the following classical result about convergence of SGD on a static distribution (see, e.g., [18]). The step size is chosen such that it matches the step size of Theorem 3.2 when $\epsilon = 0$. We include the proof for completeness.

**Lemma E.1.** *Under assumptions (A1), (A2), and (A3), lazy deploy satisfies the following:*

$$\mathbb{E}\left[\|\varphi_{k,j+1} - G(\theta_k)\|_2^2\right] \leq \left(1 - 2\eta_{k,j}\gamma + \eta_{k,j}^2 L^2\right)\mathbb{E}\left[\|\varphi_{k,j} - G(\theta_k)\|_2^2\right] + \eta_{k,j}^2\sigma^2.$$

*If, additionally, $\eta_{k,j} = \frac{1}{\gamma j + 8L^2/\gamma}$, then for all $k \geq 1, j \geq 0$, the following is true*

$$\mathbb{E}\left[\|\varphi_{k,j+1} - G(\theta_k)\|_2^2\right] \leq \frac{M_{\mathrm{lazy}}}{\gamma^2 j + L^2},$$

*where $M_{\mathrm{lazy}} \overset{\mathrm{def}}{=} \max\left\{1.2\sigma^2, 8L^2\,\mathbb{E}[\|\theta_k - G(\theta_k)\|_2^2]\right\}$.*

*Proof.* First we prove the recursion. Since $\Theta$ is closed and convex, we know

$$\mathbb{E}\left[\|\varphi_{k,j+1} - G(\theta_k)\|_2^2\right]$$

$$= \mathbb{E}\left[\left\|\Pi_\Theta\left(\varphi_{k,j} - \eta_{k,j}\nabla\ell(z_j^{(k)}; \varphi_{k,j})\right) - G(\theta_k)\right\|_2^2\right]$$

$$\leq \mathbb{E}\left[\left\|\varphi_{k,j} - \eta_{k,j}\nabla\ell(z_j^{(k)}; \varphi_{k,j}) - G(\theta_k)\right\|_2^2\right]$$

$$= \mathbb{E}\left[\|\varphi_{k,j} - G(\theta_k)\|_2^2\right] - 2\eta_{k,j}\mathbb{E}\left[\nabla\ell(z_j^{(k)}; \varphi_{k,j})^\top(\varphi_{k,j} - G(\theta_k))\right] + \eta_{k,j}^2\mathbb{E}\left[\|\nabla\ell(z_j^{(k)}; \varphi_{k,j})\|_2^2\right].$$

Next, we examine the cross-term. By the first-order optimality conditions for convex functions (Lemma B.3), we know that $\mathbb{E}\left[\nabla\ell(z_j^{(k)}; G(\theta_k))^\top(\varphi_{k,j} - G(\theta_k))\right] \geq 0$. Using this lemma along with strong convexity, we can lower bound this term as follows,

$$\mathbb{E}\left[\nabla\ell(z_j^{(k)}; \varphi_{k,j})^\top(\varphi_{k,j} - G(\theta_k))\right] \geq \mathbb{E}\left[(\nabla\ell(z_j^{(k)}; \varphi_{k,j}) - \nabla\ell(z_j^{(k)}; G(\theta_k))^\top(\varphi_{k,j} - G(\theta_k))\right]$$

$$\geq \gamma\,\mathbb{E}\left[\|\varphi_{k,j} - G(\theta_k)\|_2^2\right].$$

For the final term, we use our assumption on the second moment of the gradients,

$$\mathbb{E}\left[\|\nabla\ell(z_j^{(k)}; \varphi_{k,j})\|_2^2\right] \leq \sigma^2 + L^2\,\mathbb{E}\left[\|\varphi_{k,j} - G(\theta_k)\|_2^2\right].$$

Putting everything together, we get the desired recursion,

$$\mathbb{E}\left[\|\varphi_{k,j+1} - G(\theta_k)\|_2^2\right] \leq (1 - 2\eta_{k,j}\gamma + \eta_{k,j}^2 L^2)\,\mathbb{E}\left[\|\varphi_{k,j} - G(\theta_k)\|_2^2\right] + \eta_{k,j}^2\sigma^2.$$

Now we turn to proving the second part of the lemma. Similarly to Theorem 3.2, we prove the result using induction. As in the theorem statement, we let $\eta_{k,j} = \frac{1}{\gamma(j+k_0)}$, where we denote $k_0 = \frac{8L^2}{\gamma^2}$. The base case, $j = 0$, is trivially true by construction of the bound and choice of $k_0$. Now, we adopt the inductive hypothesis that

$$\mathbb{E}\left[\|\varphi_{k,j+1} - G(\theta_k)\|_2^2\right] \leq \frac{\max\left\{1.2\sigma^2, 8L^2\,\mathbb{E}\left[\|\theta_k - G(\theta_k)\|_2^2\right]\right\}}{\gamma^2(j+k_0)}.$$

Then, by part (a) of this lemma, it is true that

$$
\begin{aligned}
\mathbb{E}\left[\|\varphi_{k,j+2} - G(\theta_k)\|_2^2\right] &\leq \left(1 - 2\eta_{k,j}\gamma + \eta_{k,j}^2 L^2\right)\mathbb{E}\left[\|\varphi_{k,j+1} - G(\theta_k)\|_2^2\right] + \eta_{k,j}^2\sigma^2 \\
&\leq \frac{1}{\gamma^2}\left(\frac{j+k_0-2+\frac{L^2}{\gamma^2 k_0}}{(j+k_0)^2}\max\left\{1.2\sigma^2, 8L^2\,\mathbb{E}\left[\|\theta_k - G(\theta_k)\|_2^2\right]\right\} + \frac{\sigma^2}{(j+k_0)^2}\right) \\
&\leq \frac{1}{\gamma^2}\left(\frac{j+k_0-15/8}{(j+k_0)^2}\max\left\{1.2\sigma^2, 8L^2\,\mathbb{E}\left[\|\theta_k - G(\theta_k)\|_2^2\right]\right\} + \frac{\sigma^2}{(j+k_0)^2}\right) \\
&\leq \frac{1}{\gamma^2}\left(\frac{j+k_0-1}{(j+k_0)^2}\max\left\{1.2\sigma^2, 8L^2\,\mathbb{E}\left[\|\theta_k - G(\theta_k)\|_2^2\right]\right\} - \frac{7/8\cdot 1.2\sigma^2+\sigma^2}{(j+k_0)^2}\right) \\
&= \frac{1}{\gamma^2}\cdot\frac{j+k_0-1}{(j+k_0)^2}\max\left\{1.2\sigma^2, 8L^2\,\mathbb{E}\left[\|\theta_k - G(\theta_k)\|_2^2\right]\right\} \\
&\leq \frac{1}{\gamma^2}\cdot\frac{1}{j+1+k_0}\max\left\{1.2\sigma^2, 8L^2\,\mathbb{E}\left[\|\theta_k - G(\theta_k)\|_2^2\right]\right\},
\end{aligned}
$$

where the last step follows because $(j+k_0)^2 > (j+k_0)^2 - 1 = (j+k_0+1)(j+k_0-1)$. Therefore, we have shown $\mathbb{E}\left[\|\varphi_{k,j+2} - G(\theta_k)\|_2^2\right] \leq \frac{M_{\text{lazy}}}{\gamma^2(j+1+k_0)}$, which completes the proof by induction. □

### E.1  Proof of Theorem 3.3

First we state two deterministic identities used in the proof, which follow from Lemma B.4:

$$\|G(\theta) - \theta_{\text{PS}}\|_2 \leq \epsilon\frac{\beta}{\gamma}\|\theta - \theta_{\text{PS}}\|_2, \tag{2}$$

$$\|\theta - G(\theta)\|_2 \leq \|\theta - \theta_{\text{PS}}\|_2 + \|\theta_{\text{PS}} - G(\theta)\|_2 \leq \left(1 + \epsilon\frac{\gamma}{\beta}\right)\|\theta - \theta_{\text{PS}}\|_2. \tag{3}$$

Note that identity (3) implies $\|\theta - G(\theta)\|_2 < 2\|\theta - \theta_{\text{PS}}\|_2$ if $\epsilon < \frac{\gamma}{\beta}$.

By triangle inequality, we have

$$
\begin{aligned}
&\mathbb{E}\left[\|\theta_{k+1} - \theta_{\text{PS}}\|_2^2\right] \\
&= \mathbb{E}\left[\|\theta_{k+1} - G(\theta_k) + G(\theta_k) - \theta_{\text{PS}}\|_2^2\right] \\
&\leq \mathbb{E}\left[\|\theta_{k+1} - G(\theta_k)\|_2^2\right] + 2\,\mathbb{E}\left[\|\theta_{k+1} - G(\theta_k)\|_2\|G(\theta_k) - \theta_{\text{PS}}\|_2\right] + \mathbb{E}\left[\|G(\theta_k) - \theta_{\text{PS}}\|_2^2\right]. \tag{4}
\end{aligned}
$$

Denoting $k_0 = \frac{8L^2}{\gamma^2}$, Lemma E.1 bounds the first term by

$$
\begin{aligned}
\mathbb{E}\left[\|\theta_{k+1} - G(\theta_k)\|_2^2\right] &= \mathbb{E}\left[\mathbb{E}\left[\|\theta_{k+1} - G(\theta_k)\|_2^2 \mid \theta_k\right]\right] \\
&\leq \frac{1.2\sigma^2 + 8L^2\,\mathbb{E}\left[\|\theta_k - G(\theta_k)\|_2^2\right]}{\gamma^2(n(k)+k_0)} \\
&\leq \frac{1.2\sigma^2 + 32L^2\,\mathbb{E}\left[\|\theta_k - \theta_{\text{PS}}\|_2^2\right]}{\gamma^2(n(k)+k_0)},
\end{aligned}
$$

where in the last step we apply identity (3). Note also that by Jensen's inequality, we know

$$\mathbb{E}\left[\|\theta_{k+1} - G(\theta_k)\|_2\right] \leq \frac{1.1\sigma + 6L\,\mathbb{E}\left[\|\theta_k - G(\theta_k)\|_2\right]}{\gamma\sqrt{n(k)+k_0}}.$$

We can use this inequality, together with identities (2) and (3), to bound the cross-term in equation (4) as follows:

$$2\,\mathbb{E}\left[\|\theta_{k+1} - G(\theta_k)\|_2 \|G(\theta_k) - \theta_{\mathrm{PS}}\|_2\right]$$

$$\leq 2\epsilon\frac{\beta}{\gamma}\,\mathbb{E}\left[\|\theta_{k+1} - G(\theta_k)\|_2 \|\theta_k - \theta_{\mathrm{PS}}\|_2\right]$$

$$\leq \frac{2\epsilon\frac{\beta}{\gamma}}{\sqrt{n(k)+k_0}}\,\mathbb{E}\left[\left(\frac{6L}{\gamma}\|\theta_k - G(\theta_k)\|_2 + \frac{1.1\sigma}{\gamma}\right)\|\theta_k - \theta_{\mathrm{PS}}\|_2\right]$$

$$\leq \frac{2\epsilon\frac{\beta}{\gamma}}{\sqrt{n(k)+k_0}}\,\mathbb{E}\left[\left(\frac{6L}{\gamma}\left(1 + \epsilon\frac{\beta}{\gamma}\right)\|\theta_k - \theta_{\mathrm{PS}}\|_2 + \frac{1.1\sigma}{\gamma}\right)\|\theta_k - \theta_{\mathrm{PS}}\|_2\right]$$

$$\leq \frac{24\epsilon\beta L}{\gamma^2\sqrt{n(k)+k_0}}\,\mathbb{E}\left[\|\theta_k - \theta_{\mathrm{PS}}\|_2^2\right] + \frac{2.2\sigma\epsilon\beta}{\gamma^2\sqrt{n(k)+k_0}}\,\mathbb{E}\left[\|\theta_k - \theta_{\mathrm{PS}}\|_2\right].$$

We bound the latter term by applying the AM-GM inequality; in particular, for all $\alpha_0 \in (0,1)$, it holds that

$$\frac{2.2\sigma\epsilon\beta}{\gamma^2\sqrt{n(k)+k_0}}\,\mathbb{E}\left[\|\theta_k - \theta_{\mathrm{PS}}\|_2\right] \leq \frac{1.1\sigma\epsilon\beta}{\gamma^2}\left(\frac{1}{(n(k)+k_0)^{\alpha_0}} + \frac{\mathbb{E}\left[\|\theta_k - \theta_{\mathrm{PS}}\|_2^2\right]}{(n(k)+k_0)^{1-\alpha_0}}\right).$$

Thus, the final bound on the cross-term in equation (4) is

$$2\,\mathbb{E}\left[\|\theta_{k+1} - G(\theta_k)\|_2 \|G(\theta_k) - \theta_{\mathrm{PS}}\|_2\right] \leq \left(\frac{24\epsilon\beta L}{\gamma^2\sqrt{n(k)+k_0}} + \frac{1.1\sigma\epsilon\beta}{\gamma^2(n(k)+k_0)^{1-\alpha_0}}\right)\mathbb{E}\left[\|\theta_k - \theta_{\mathrm{PS}}\|_2^2\right]$$
$$+ \frac{1.1\sigma\epsilon\beta}{\gamma^2(n(k)+k_0)^{\alpha_0}}.$$

The final term in equation (4) can be bounded by identity (2):

$$\mathbb{E}\left[\|G(\theta_k) - \theta_{\mathrm{PS}}\|_2^2\right] \leq \left(\epsilon\frac{\beta}{\gamma}\right)^2 \mathbb{E}\left[\|\theta_k - \theta_{\mathrm{PS}}\|_2^2\right].$$

Putting all the steps together, we have derived the following recursion, true for all $\alpha_0 \in (0,1)$:

$$\mathbb{E}\left[\|\theta_{k+1} - \theta_{\mathrm{PS}}\|_2^2\right] \leq \left(\frac{32L^2}{\gamma^2(n(k)+k_0)} + \frac{24\epsilon\beta L}{\gamma^2\sqrt{n(k)+k_0}} + \frac{1.1\sigma\epsilon\beta}{\gamma^2(n(k)+k_0)^{1-\alpha_0}} + \left(\epsilon\frac{\beta}{\gamma}\right)^2\right)\mathbb{E}\left[\|\theta_k - \theta_{\mathrm{PS}}\|_2^2\right]$$
$$+ \frac{1.2\sigma^2}{\gamma^2(n(k)+k_0)} + \frac{1.1\sigma\epsilon\beta}{\gamma^2(n(k)+k_0)^{\alpha_0}}$$
$$\leq c\,\mathbb{E}\left[\|\theta_k - \theta_{\mathrm{PS}}\|_2^2\right] + \frac{1.2\sigma^2}{\gamma^2(n(k)+k_0)} + \frac{1.1\sigma\epsilon\beta}{\gamma^2(n(k)+k_0)^{\alpha_0}}, \tag{5}$$

where we define

$$c \overset{\text{def}}{=} \frac{32L^2}{\gamma^2 n_0} + \frac{24\epsilon\beta L}{\gamma^2\sqrt{n_0}} + \frac{1.1\sigma\epsilon\beta}{\gamma^2 n_0^{1-\alpha_0}} + \left(\epsilon\frac{\beta}{\gamma}\right)^2. \tag{6}$$

We pick $n_0$ large enough such that there exists $\alpha_0 > 0$ for which $c < 1$.

Unrolling the recursion given by equation (5), we get

$$\mathbb{E}\left[\|\theta_{k+1} - \theta_{\mathrm{PS}}\|_2^2\right] \leq c^k \|\theta_1 - \theta_{\mathrm{PS}}\|_2^2 + \frac{1}{\gamma^2}\sum_{j=1}^{k} c^{k-j}\left(\frac{1.2\sigma^2}{n(j)+k_0} + \frac{1.1\sigma\epsilon\beta}{(n(j)+k_0)^{\alpha_0}}\right).$$

Since $\alpha_0 < 1$, we can upper bound the second term as

$$\frac{1}{\gamma^2}\sum_{j=1}^{k} c^{k-j}\left(\frac{1.2\sigma^2}{n(j)+k_0} + \frac{1.1\sigma\epsilon\beta}{(n(j)+k_0)^{\alpha_0}}\right)$$

$$\leq \frac{1.2\sigma^2}{\gamma^2}\sum_{j=1}^{k} c^{k-j}\frac{1}{n(j)+k_0} + \frac{1.1\sigma\epsilon\beta}{\gamma^2}\sum_{j=1}^{k} c^{k-j}\frac{1}{(n(j)+k_0)^{\alpha_0}}$$

$$\leq \frac{1}{\gamma^2(1-c)}\left(\frac{1.2\sigma^2}{n_0}(2k^{-\alpha} + c^{(1-2^{-1/\alpha})k}) + \frac{1.1\sigma\epsilon\beta}{n_0^{\alpha_0}}(2k^{-\alpha\cdot\alpha_0} + c^{(1-2^{-1/(\alpha\alpha_0)})k})\right)$$

where in the second inequality we apply Lemma B.5 after plugging in the choice of $n(k)$. Using the fact that $\alpha_0 \in (0,1)$ and hence $c^{(1-2^{-1/(\alpha\alpha_0)})k} < c^{(1-2^{-1/\alpha})k}$, as well as $\epsilon < \frac{\gamma}{\beta}$ and $n_0 \geq 1$, gives

$$
\frac{1}{\gamma^2(1-c)} \left( \frac{1.2\sigma^2}{n_0}(2k^{-\alpha} + c^{(1-2^{-1/\alpha})k}) + \frac{1.1\sigma\epsilon\beta}{n_0^{\alpha_0}}(2k^{-\alpha\cdot\alpha_0} + c^{(1-2^{-1/(\alpha\alpha_0)})k}) \right)
$$

$$
\leq \frac{1.2\sigma^2 + 1.1\sigma\gamma}{\gamma^2(1-c)} \left( 4k^{-\alpha\alpha_0} + 2c^{(1-2^{-1/\alpha})k} \right)
$$

$$
\leq \frac{3(\sigma+\gamma)^2}{\gamma^2(1-c)} \left( 2k^{-\alpha\alpha_0} + c^{\Omega(k)} \right).
$$

It remains to set $\alpha_0$; we set $\alpha_0 = \max\{\delta \in (0,1) : c < 1\}$ (note that the existence of such $\alpha_0$ is guaranteed by the choice of $n_0$). Clearly, $\alpha_0 \to 1$ as $n_0$ grows, and so putting everything together gives

$$
\mathbb{E}\left[\|\theta_{k+1} - \theta_{\text{PS}}\|_2^2\right] \leq c^k \|\theta_1 - \theta_{\text{PS}}\|_2^2 + \frac{3(\sigma+\gamma)^2}{\gamma^2(1-c)} \left( \frac{2}{k^{\alpha\cdot(1-o(1))}} + c^{\Omega(k)} \right),
$$

as desired.

## F   Proof of Corollary 3.4

From Theorem 3.2, we know that for greedy deploy, $\mathbb{E}\left[\|\theta_{k+1} - \theta_{\text{PS}}\|_2^2\right] = \mathcal{O}(\frac{1}{k})$ where $k$ indexes both the number of classifiers and the number of samples collected. By inverting this bound, we see that to ensure $\mathbb{E}\left[\|\theta_{k+1} - \theta_{\text{PS}}\|_2^2\right] \leq \delta$, it suffices to collect $\mathcal{O}(\frac{1}{\delta})$ samples.

From our analogous convergence result for lazy deploy (Theorem 3.3), we know that after the $k$-th deployment, it holds that $\mathbb{E}\left[\|\theta_{k+1} - \theta_{\text{PS}}\|_2^2\right] = \mathcal{O}(1/k^{\alpha\cdot\omega})$, for some $\omega = 1 - o(1)$ which is independent of $k$ and tends to 1 as $n_0$ grows. If we collect $\Theta(j^\alpha)$ samples for each deployment $j = 1 \ldots k$, after $k$ deployments the total number of samples $N$ is $\Theta(k^{\alpha+1})$. Therefore,

$$
\mathbb{E}\left[\|\theta_{k+1} - \theta_{\text{PS}}\|_2^2\right] = \mathcal{O}(1 \,/\, N^{\frac{\alpha\cdot\omega}{\alpha+1}}).
$$

By inverting these bounds, we get our desired result for the asymptotics of lazy deploy.