[Reviews · NeurIPS 2020]

Review 1

Summary and Contributions: This work considers the problem of performative prediction in stochastic setting - where the current model influences the data distribution which is observed in the future due to deployment. The broad task of performative prediction is to minimize losses while also accounting for the fact that new policies change the data distribution. In particular, this work is concerned with finding a `stable’ predictor which when deployed, also minimizes the loss with respect to the data induced by itself. This work gives the first precise analysis of stochastic gradient methods to this problem whenever the sensitivity is small enough compared to the inverse condition number. Two regimes are considered:
 1. Greedy deployment where each stochastic gradient step is used to change the model. 2. Lazy deployment where multiple i.i.d samples are collected from the current model before updating the parameters. In the greedy deployment regime, O(1/k) convergence rates are established whereas in the lazy regime O(1/k^{alpha}) rates are established for arbitrary alpha.

Strengths: I think the work is foundational and begins the study of a fundamental problem which has a lot of applications. The results are precise and succinct and give sharp criterion for convergence of gradient based methods. The paper is well written and easy to understand from a technical stand point and the simulations are really helpful. Stochastic case is the most important one in such problems since this is what one encounters in practice. This topic is very relevant to NeurIPS community in particular and Machine Learning community in general. ======== The authors have addressed most of my concerns with the rebuttal. I maintain my score.

Weaknesses: Given the foundational nature of the work, I feel it would be better if some examples of the models are described in the main body of the paper - along with some intuition regarding performative stability using a concrete running example.
 I feel that most readers have not seen many works on this topic and lack intuition regarding the fundamental quantities of interest considered here. For instance, why are stable distributions the right thing to look at as opposed to playing a cat-and-mouse chase of distributions which reduces `regret' given by the losses (for instance, in a natural example) ? A minor quib: Is it possible to set the stability parameter epsilon to be small enough by scaling \theta? This shouldn’t change the set of available models at all. Scaling also shouldn’t affect the condition number even though it affects the smoothness and strong convexity parameter.

Correctness: I checked and found the main proofs to be correct to the best of my knowledge.

Clarity: I have answered this in the strengths/weaknesses section.

Relation to Prior Work: Apart from this work, to the best of my knowledge, there have been very few works on this particular problem and the authors cite all of them. On a high level this seems very similar to the fundamental problem in reinforcement learning. A paragraph or two explaining the connections or differences would help the readers appreciate this problem in a better context.

Reproducibility: Yes

Additional Feedback:


Review 2

Summary and Contributions: This paper uses stochastic optimization to study performative prediction and proves non-asymptotic rates of convergence for two variants of the stochastic gradient method (i.e., greedy deploy and lazy deploy) under certain conditions.

Strengths: This paper presents that different time intervals at which they deploy models trained with stochastic gradient descent in performative prediction leads to qualitatively different algorithms. Moreover, a series of experimental results confirm the result of theoretical analysis of greedy and lazy deploy.

Weaknesses: Although the paper is theoretically sound, there are still some questions need to be discussed in this paper: 1. About the innovation. This paper uses stochastic gradient methods to solve performative prediction, which is similar to the previous work [R1]. In addition, the proof in this paper seems an extension of the prior work [R1] on performative prediction. The authors should give the key contribution of this paper and discuss the difference between them. [R1] Juan C. Perdomo, Tijana Zrnic, Celestine Mendler-Dünner, and Moritz Hardt. Performative Prediction. In Proceedings of the International Conference on Machine Learning (ICML), 2020. 2. About the experiment. The authors only report the experimental results of the proposed algorithms, and do not compare their experimental results with related methods such as [R1]. Thus, I suggest that the authors design some new experiments to demonstrate the performance of the proposed methods.

Correctness: The paper is theoretically sound. Moreover, a series of experimental results confirm the result of theoretical analysis of greedy and lazy deploy.

Clarity: The paper is well structured and the writing is clear overall.

Relation to Prior Work: The paper is well structured and the writing is clear overall.

Reproducibility: Yes

Additional Feedback: This paper presents that different time intervals at which they deploy models trained with stochastic gradient descent in performative prediction leads to qualitatively different algorithms. Moreover, a series of experimental results confirm the result of theoretical analysis of greedy and lazy deploy. Although the paper is theoretically sound, there are still some questions need to be discussed in this paper: 1. About the innovation. This paper uses stochastic gradient methods to solve performative prediction, which is similar to the previous work [R1]. In addition, the proof in this paper seems an extension of the prior work [R1] on performative prediction. The authors should give the key contribution of this paper and discuss the difference between them. [R1] Juan C. Perdomo, Tijana Zrnic, Celestine Mendler-Dünner, and Moritz Hardt. Performative Prediction. In Proceedings of the International Conference on Machine Learning (ICML), 2020. 2. About the experiment. The authors only report the experimental results of the proposed algorithms, and do not compare their experimental results with related methods such as [R1]. Thus, I suggest that the authors design some new experiments to demonstrate the performance of the proposed methods. ###################### Post author response ################## In the responses, most of my concerns have been addressed. In particular, my main concern is that the main contributions of this paper are similar to one ICML2020 paper. I am also satisfied with the author feedback. I agree that the theoretical result of the paper is important. So, I am also fine for accepting the paper.


Review 3

Summary and Contributions: The paper addresses the stochastic optimization for performative prediction, which is proposed recently by Perdomo et al. The main contribution of the paper: providing non-asymptotic convergence guarantees for both greedy deploy and lazy deploy in performative settings. *********************** I read other review reports and authors feedback. The authors address all of my questions. I will keep previous rating of this paper. ***********************

Strengths: The theoretical results: proving convergence guarantees for greedy and lazy deploy algorithms. Theorem 3.2 and 3.3 show that both algorithms are able to converge to stability and provide an insight about the convergence speed for each of the algorithms. Based on the main theorems, the paper also suggests how to choose between greedy and lazy deploy models, based on the strength of performative effects, which is presented in corollary 3.4. This result is practically useful as in certain contexts, the sample size and the required precision are different for each algorithm. To illustrate the theoretical analysis, the paper provides experiments on synthetic data, which is a simple Gaussian model, and strategic classification, using a credit scoring dataset. The experimental results are consistent with the theoretical results. I believe the contribution of the paper is novel and of interest in ML community.

Weaknesses: I could not find any significant flaw that affects the contribution of this work. I am looking forward to reading other reviews.

Correctness: The claims and method are correct as well as the empirical methodology used in the paper.

Clarity: The paper is well written. All the explanations and claims are clear. Due to the page limit, the proofs and the experimental results are presented in detail in supplementary.

Relation to Prior Work: The paper clearly discuss the difference between its work and previous contributions. As performative prediction has only been studied recently, the work of the paper is unique.

Reproducibility: Yes

Additional Feedback: The results in the paper are established in strong convexity setting. Since most of machine learning problems are non-convex, could it be possible that we can relax the convexity condition and still obtain convergence result for greedy and lazy deploy algorithms?


Review 4

Summary and Contributions: The paper studies stochastic optimization for performative prediction, and proves non-asymptotic rates of convergence for greedily deploying models after each stochastic update or for taking several updates before redeploying.

Strengths: This paper establishes strong theoretical understandings, which could be of fundamental importance for stochastic optimization.

Weaknesses: N/A.

Correctness: Based on limited reading, the results are technically sound.

Clarity: The paper is written clearly.

Relation to Prior Work: The relation to prior work was carefully explained in the paper.

Reproducibility: Yes

Additional Feedback:

[Author Response · NeurIPS 2020]

# Author Response: Stochastic Optimization for Performative Prediction – Paper #28

We thank the reviewers for their feedback and look forward to incorporating these comments into our revised manuscript. We address below the remaining questions and comments.

## Reviewer 1

**Further intuition & examples.** We appreciate the suggestion of including an additional example in the main body of the paper illustrating performative stability as well as the behavior of lazy/greedy deploy on this example. We agree that this would aid the reader who is unfamiliar with the framework, and will think about a good example to incorporate.

We see two main reasons why stable solutions are desirable. First, in most real-world systems frequent retraining comes at a significant cost; stability, on the other hand, removes the need for retraining. Second, once the distribution shifts as a response to model deployment, we in general have no guarantees as to the performance of the deployed model. Performative stability ensures the model will have nearly optimal predictive power, as alluded to in L:130-136.

**Connection to reinforcement learning (RL).** The discussion section in [15] analyzes connections between performative prediction and RL in detail. One way of understanding performative prediction is as a particular case of a reinforcement learning problem with special structure ($\epsilon$-sensitivity, restricted reward functions) that makes it tractable. To provide more context we will elaborate on these connections in our revised version, and additionally discuss how our ideas connect with the stochastic optimization literature in RL.

**Scale-invariance of the sensitivity parameter.** Thank you for the careful observation, we will clarify this. Namely, it is not possible to reduce sensitivity by scaling the parameter $\theta$. The reason is that the notion of *joint* smoothness we consider does not scale like strong convexity with the rescaling of $\theta$. For example, rescaling $\theta \mapsto 2\theta$ (thus making $\epsilon \mapsto \epsilon/2$) would downscale the strong convexity parameter and the parameter corresponding to the usual notion of smoothness in optimization by 4, however the smoothness in $z$ (second inequality in L:143) would downscale by 2. Therefore, the critical ratio necessary for convergence, $\epsilon \frac{\beta}{\gamma} < 1$, is unaltered by scaling.

## Reviewer 2

**Contribution over prior work.** As outlined in the introduction and later emphasized in L:88-91, our work does build on the framework of performative prediction introduced in [15]. In [15], the authors largely focus on proving convergence of repeated risk minimization/gradient descent in settings where the learner has access to the *full* distribution. While they provide an extension of their results to the finite-sample regime, their results in this regime are quite weak since their analysis relies on concentration of the empirical distribution to the true distribution in the Wasserstein metric. As a result, they require the learner to collect *exponentially* many samples in the dimension at every step (and, in fact, their rate is only asymptotic).

In contrast, our analysis ensures convergence even if the learner collects a *single* sample at every step, something that is not at all guaranteed in [15]. To achieve this result, we rely on a different proof technique and a fundamentally new analysis of the stochastic gradient method in performative contexts. We will extend this comparison in the related work section to make the technical novelty of our proofs more clear.

**Experimental comparison to prior work.** We hope that the above discussion clarifies why a comparison to [15] is not meaningful. All experiments in [15] evaluate convergence when the learner has access to the full distribution at every step, while we carry out experiments in a finite-sample setting. Indeed, 4/5 of our plots have the number of samples collected on the x-axis, a quantity that is not well-defined within the experimental setup of [15].

That said, we will happily carry out a new experiment at the population level in Figure 3b to illustrate to the reader the slowdown in convergence rate caused by the stochastic (versus exact) nature of the gradient updates.

## Reviewer 3

**Extension to non-convex setting.** We believe that extending our results to non-convex settings is an important and exciting direction for future work. In Proposition 2.4 we show that repeated gradient descent (essentially the population-level analog of greedy deploy) need not converge even for weakly convex losses — thus, studying non-convex settings through the lens of performative prediction would likely require a completely new set of algorithmic tools.

## Reviewer 4

We thank the reviewer for the positive assessment of our work.

[Meta-Review · NeurIPS 2020]

Reviewers had agreement that this is a good contribution to stochastic optimization research. The concerns with respect to the similarity with a recent ICML paper has been addressed in the author rebuttal.